# Terrestrial laser scanning for quantifying small-scale vertical movements of the ground surface in Artic permafrost regions

Sabrina Marx<sup>1</sup>, Katharina Anders<sup>1</sup>, Sofia Antonova<sup>2</sup>, Inga Beck<sup>1</sup>, Julia Boike<sup>2</sup>, Philip Marsh<sup>3</sup>, Moritz Langer<sup>2,4</sup>, Bernhard Höfle<sup>1,5</sup>

<sup>1</sup> GIScience, Institute of Geography, Heidelberg University, Heidelberg, 69120 Germany
 <sup>2</sup> Alfred Wegener Institute, Helmholtz Center for Polar and Marine Research, Potsdam, 14473, Germany
 <sup>3</sup> Cold Regions Research Centre, Wilfrid Laurier University, Waterloo, N2L 3C5, ON, Canada
 <sup>4</sup> Department of Geography, Humboldt-University, Berlin, 10099, Germany
 <sup>5</sup> Heidelberg Center for the Environment (HCE), Heidelberg University, Heidelberg, 69120, Germany

Correspondence to: Sabrina Marx (marx@uni-heidelberg.de)

**Abstract.** Three-dimensional data acquired by terrestrial laser scanning (TLS) provides an accurate representation of Earth's surface, which is commonly used to detect and quantify topographic changes on a small scale. However, in Arctic permafrost regions the tundra vegetation and the micro-topography have significant effects on the surface representation in the captured dataset. The resulting spatial sampling of the ground is never identical between two TLS surveys. Thus, monitoring of heave

- and subsidence in the context of permafrost processes are challenging. This study evaluates TLS for quantifying small-scale vertical movements in an area located within the continuous permafrost zone, 50 km north-east of Inuvik, Northwest Territories, Canada. We propose a novel filter strategy, which accounts for spatial sampling effects and identifies TLS points suitable for multi-temporal deformation analyses. Further important prerequisites must be met, such as accurate co-registration of the TLS datasets. We found that if the ground surface is captured by more than one TLS scan position, plausible subsidence
- rates (up to mm-scale) can be derived; compared to e.g. standard raster-based DEM difference maps which contain change rates strongly affected by sampling effects.

## **1** Introduction

Approximately one quarter of Northern hemisphere lands contain permafrost, which is defined as ground material that remains at or below 0 °C during at least two consecutive years. Its depth varies from centimetres to hundreds of metres and reaches up

- to 1500 m below the land surface in particular areas of Eastern Siberia (Melnikov, 1967). Large quantities of organic carbon are preserved in the permafrost, an amount that is likely twice as large as the current quantity of carbon in the atmosphere (Zimov et al., 2006). Current global climate warming is more intense in high latitudes (IPCC, 2013) and causes permafrost to warm and thaw, and, consequently, to release the carbon into atmosphere, amplifying global warming. Lowering of land surface due to permafrost thaw can manifest in various ways, including thermokarst, thaw slumps, and slope failures (Rowland
- et al., 2010). Besides these spatially confined processes, relatively uniform and gradual isotropic thaw subsidence, which is not exhibited by any surface disturbance, was recently observed in Arctic permafrost regions (e.g. Shiklomanov et al., 2013).

In-situ measurements on such isotropic subsidence are often performed using differential global positioning system (DGPS) measurements (e.g. Little et al., 2003; Shiklomanov et al., 2013; Beck et al., 2015; Streletskiy et al., 2016) or thaw-tubes (e.g. Nixon et al., 2003; Short et al., 2014). However, such discrete point measurements do not allow a detailed assessment of surface deformations. As a result, remote sensing techniques have been increasingly exploited to provide data in a more

- comprehensive spatial coverage as well as higher spatial and temporal resolution (Arenson et al., 2016). For example, Günther et al. (2015) used multi-temporal Digital Elevation Models (DEM), obtained from aerial and satellite imagery, and found an average net subsidence of 3.6 ±1.8 m over 62 years in an ice-rich permafrost area of Siberian Arctic. Differential Synthetic Aperture Radar interferometry (DInSAR) is another remote-sensing technique which recently has been adapted for monitoring permafrost thaw subsidence (e.g. Liu et al., 2010; Short et al., 2011). However, the usage of DInSAR can be limited due to
- interferometric phase decorrelation (e.g. Beck et al., 2015; Antonova et al., 2016) as well as atmospheric effects (e.g. Short et al., 2011). Bhardwaj et al. (2016) identifies Light detection and ranging (LiDAR), also referred to as laser scanning (Höfle and Rutzinger, 2011), as potentially being the best tool for increasing the performance of measuring permafrost-related processes, such as mass movements, vegetation dynamics and topographical characteristics (Bhardwaj et al., 2016). By combining point measurements with airborne laser scanning (ALS), and geophysical datasets, the study of Hubbard et al. (2013) indicates a
- close linkage between micro-topography, active layer, and permafrost variability. This linkage of subsurface and land surface variabilities emphasizes the potential of laser scanning to indirectly characterize subsurface properties in permafrost environments in a non-invasive manner (Hubbard et al., 2013).

In an extensive review, Bhardwaj et al. (2016) discuss the application of laser scanning for monitoring permafrost processes. Terrestrial laser scanning (TLS) was first used in high-mountain environments for the study of rock glaciers (Avian et al.,

- 2009, 2008; Bodin et al., 2008; Deline et al., 2008). As the accuracy of laser scanning systems has improved, ALS and TLS have been used to study lowland sub-Arctic permafrost regions (Chasmer et al., 2011; Gangodagamage et al., 2014; Wainwright et al., 2015; Nouwakpo et al., 2016). In this context, the authors point out that the small number of studies using laser scanning to consider changes in permafrost is still insufficient for an improved understanding of these frozen ground areas. To date, laser scanning data, has been primarily acquired by airborne systems in order to derive DEMs, which are
- analyzed coupled to remote sensing imagery and/or in situ measurements (e.g. Jones et al., 2015). Such ALS data typically has decimeter resolution (Barnhart and Crosby, 2013), and are not sufficiently accurate to study seasonal vertical movements of the ground caused by thawing and freezing processes of the permafrost which is usually in the order of a few mm to cm. TLS monitoring is required to study such changes.

For TLS-based change detection and deformation analysis, one has to consider the following four main challenges: (1) local

variations in measurement geometry and surface properties, which means that parts of the surface might not be sampled in a repeated TLS survey, (2) registration errors, which can result in systematic shifts, (3) different viewpoints, which may lead to occlusion effects and (4) temporary objects (Lindenbergh and Pietrzyk, 2015). The sampling effect (challenge 1), is caused by various factors, such as vegetation height and density, as well as surface roughness and the scan setup. Due to occlusion effects (challenge 3) by solid objects the laser beam may not hit the ground at every point and, thus, the spatial sampling of the surface

of interest is never identical between two surveys (Lague et al., 2013). Furthermore, the monitoring setup with its incidence angles directly affecting penetration depths has an influence on the detectability of small-scale subsidence rates. The mentioned effects are relevant for quantifying subsidence in permafrost areas, which are dominated by e.g. organic matter, moss and shrubs and do not provide a stable ground surface. The highly dynamic nature of permafrost regions caused by freezing and

- thawing of the upper layer of the ground, is also challenging for the co-registration between two TLS surveys (challenge 2). The effect of elevation errors due to short ground vegetation is further described by Fan et al. (2014). An error of about 65% of the grass height is reported. Lower incidence angles, i.e. the angle between surface normal and laser beam, for closer ground areas mean that it is easier for laser beams to penetrate through the vegetation to a lower level and to reach the ground surface below vegetation (Fan et al., 2014). Furthermore, Fan et al. (2014) conclude that (1) lower incidence angles (greater visibility)
- reduce the occlusion effects, (2) scanning the same area from multiple locations reduces the vegetation error and (3) greater vegetation errors occur where vegetation is higher/denser. The effect of unfavorable scan geometries on soil erosion measurements is described by Eltner and Baumgart (2015). In a follow-up study, the authors utilize DEMs captured with an Unmanned Aerial System (UAS) to detect systematic errors in the TLS data and apply a selection to the point clouds with regards to scan geometry and surface roughness (Eltner et al., 2016). To the authors' knowledge, only a few studies have
- evaluated the applicability of TLS for change detection in permafrost regions (Avian et al., 2008; Kociuba et al., 2014). Indepth analyses of the effect of vegetation, micro-topography and scan setup on TLS-based subsidence measurements are lacking.

With respect to deformation analyses in general, computing DEM difference rasters is a straightforward and commonly applied method to detect topographic surface changes (Abellán et al., 2009; Goodwin et al., 2016). Thereby, the 3D point cloud is

- converted into a 2.5D representation of the surface which results in a loss of information (Lague et al., 2013). In recent years, calculating the differences directly within the point cloud has been gaining more and more attention. Barnhart and Crosby (2013) use cloud to mesh comparison (C2M) as well as Multiscale Model to Model Cloud Comparison (M3C2) developed by Lague et al. (2013) to detect surface changes on an evolving thermokarst in Alaska. They found that the M3C2 method is better suited to detect small-scale changes (down to 2-7 cm) as the distance calculation algorithms accounts for position uncertainties,
- i.e. random errors, due to averaging of point positions. Difference is then calculated along the local surface normal (Lague et al., 2013). Kromer et al. (2015) apply a modified M3C2 algorithm and relies on spatial and temporal averaging to detect displacements at sub-mm scale. In a follow-up study a near-real-time TLS monitoring system was evaluated for landslide monitoring (Kromer et al., 2017). However, TLS datasets with high temporal and spatial resolution are required for their presented 4D filtering approach, which is challenging in permafrost regions due to ground instability and extreme
- meteorological conditions in these areas (Kociuba et al., 2014). The overall aim of this study is to evaluate TLS for deformation analyses in permafrost regions by comparing a raster-based and a point-cloud based approach with particular emphasis on the effect of tundra vegetation and micro-topography on cm-scale subsidence monitoring. To account for sampling and signal occlusion effects, a new point-based filter strategy for multi-temporal deformation analyses is introduced. Further in-depth insights for TLS-based subsidence monitoring are reached by analyzing datasets of two different scan setups (repeating the

data acquisition from similar vs. different scan position). Finally, results are assessed based on a comparison to gold-standard manual subsidence measurements and on real-time kinematic (RTK) Global Navigation Satellite System (GNSS) measurements.

#### 2 Study area, field measurements and sensors

- Field work was carried out at the Trail Valley Creek (TVC) research watershed (68° 44' 17" N 133° 26' 26" W), located about 50 km North East of Inuvik, Northwest Territories, Canada (Fig. 1). TVC lies within the zone of continuous permafrost at the northern edge of the forest-tundra boundary region. Dominating vegetation is a mix of willow shrub (*Salix*) and herbaceous tundra with lichen (*Lecidea*) and mosses (*Sphagnum*) (Corns, 1974; Burn and Kokelj, 2009). The climate within this region is characterized by a strong seasonality with summer temperatures often above 20 °C, and winter temperatures below -40 °C. At
- the Inuvik, Meteorological Survey of Canada weather station, the average air temperature between the years 1971 and 2000 was about -9 °C with a mean annual precipitation of 140 mm (Burn and Kokelj, 2009). For this study, two sites within TVC, each with different vegetation density, have been investigated: Site 1 is about 50x40 m, almost flat and covered by low tundra vegetation and Site 2 is equal in size but contains more shrubs (Fig. 1).
- High-resolution point cloud data in both test sites was captured using a Riegl VZ-400 TLS on June and August 2015 and
  August 2016. This time-of-flight 3D laser scanner measures the distance between sensor and object, and calculates 3D coordinates (x, y, z) for the captured object points. The resulting point clouds have a nominal point spacing of 3 mm at 10 m distance with a range accuracy of 5 mm at 100 m scanning range (RIEGL, 2014).

Within the two study sites, measurement of 43 ground validation points is repeated during each survey period using a highend RTK GNSS (Leica Viva GS15 receiver with GS10 base station). RTK is a relative positioning technique computing xyz-

- coordinates with a vertical accuracy of 15 mm and a horizontal one of 8 mm according to the manufacturer's datasheet (Leica Geosystems, 2015). These data will be used to evaluate TLS-based subsidence measurements (Brown et al., 2000). To prevent the RTK rover pole from sinking into the active layer and ensure comparable GNSS measurements on the moss and lichen covered soil, the rover setup is placed on a board made out of high pressure laminate. The validation points are marked so that the exact same positions were measured during each survey. Additionally, 24 carbon fiber rods for manual measurement of
- subsidence are installed. The rods are drilled about 1.8 m deep into the permafrost to prevent any movement induced by heave and subsidence as well as cryoturbation. With this setup, the so-called subsidence stations serve as absolute fixed points in the otherwise highly dynamic permafrost environment. Vertical movement of the ground surface is quantified by manually measuring the protruding part of the rods. Five measurements at each subsidence station are taken to account for microtopographic variations around the rod. Subsidence per station is assessed as median value of measurements, respectively.
- However, it needs to be considered that measurements might not be consistent between the three survey dates as, for example, measurements were gathered by different persons and, though best practices are applied, recordings might vary slightly. As fixed points, the top of the carbon rods additionally serve to ensure high co-registration accuracy between the TLS datasets.

The three datasets from TLS, GNSS, and manual subsidence stations were captured during three field campaigns. For measuring seasonal subsidence rates, the first campaign took place at the beginning of the thawing period in June 2015 (2015/06/10 and 2015/06/11), the second campaign at the end of summer, in August 2015 (2015/08/21 and 2015/08/22). The following year, data acquisition was repeated in a third campaign in August 2016 (2016/08/23 and 2016/08/24), which enables us to also quantify the change record of one entire year. Based on the captured datasets, deformation analysis is performed for


the following three survey periods: 2015/06-2015/08, 2015/06-2016/08 and 2015/08-2016/08.

#### 3 Methods

In this section we present and evaluate a method to extract subsidence rates in permafrost regions using TLS. Figure 2 shows the entire processing and analysis steps starting with data acquisition, co-registration of the single scan campaigns and pre-

processing of the derived point cloud. The main focus is on the deformation analysis using raster-based and point-based approaches to quantify small-scale vertical movements of the ground surface. To account for point sampling effects, filter strategies are applied before distances between the point clouds are calculated to derive deformation information.

#### 3.1 TLS data acquisition

The two study sites are captured from seven scan positions during each TLS campaign (2015/06, 2015/08 and 2016/08). As

- 15 the scan setup, i.e. the incidence angle of the laser beam, has an impact on occlusion patterns (Fan et al., 2014; Lindenbergh and Pietrzyk, 2015), our method is evaluated for two different approaches: (1) TLS scans are repeated from the same scan positions (2015/08-2016/08) and, thus, the ground surface is scanned with similar incidence angles and (2) the surface is captured from different scan positions compared to the TLS survey before (2015/06-2015/08; 2015/06-2016/08). It has to be noted that with a non-stationary TLS setup it is not feasible to locate the laser scanner at the exact same position and height.
- 20 This results in a mean 3D distance between scan positions from 2015/08 and 2016/08 of about 20 cm (min. 14 cm, max. 29 cm).

#### 3.2 Registration and co-registration

First, the TLS point clouds are registered and georeferenced in the RiSCAN PRO software (RIEGL). The single scans of each survey are registered based on four cylindrical reflectors that are placed at the edges of the study sites. 3D distances between

the single, registered scan positions are on mm-scale at the reflector tie points (Appendix 1). Then, the registered point cloud is georeferenced into a common global coordinates system (WGS 1984, UTM 8N) using the reflectors' coordinates captured via the RTK GNSS system.

To achieve a highly accurate registration of the three different TLS campaigns, subsequent co-registration of the georeferenced point clouds is performed based on the stable subsidence stations. Five to seven coordinate pairs (top of the rods) are extracted

for each study site and used to calculate a rigid 3D transformation matrix which maintains the relative point positions. Quality

of co-registration is evaluated based on 10 control coordinate pairs: Co-registration reduces the RMSE of the z-distances between the TLS surveys to 0.2 - 0.3 cm at the control points (Table 1). Uncertainties for the TLS-derived change maps are assumed to be in the magnitude of the co-registration error in z-direction (Orem and Pelletier, 2015). Thus, prerequisites are met for quantifying small-scale vertical changes within our study sites.

## 5 3.3 Pre-processing

Pre-processing of the TLS datasets includes filtering of the datasets to a maximum range of 30 m around the scan positions in order to avoid alignment errors that increase with large distance to the respective scan position. Furthermore, spatially isolated points are removed via a statistical outlier filter (settings: number of neighbors, 10; standard deviation multiplier threshold, 1; Rusu and Cousins, 2011). Point clouds are then further processed using OPALS (version 2.1.5; Pfeifer et al., 2014), Point Cloud Library (PCL) (Rusu and Cousins, 2011) and in CloudCompare (version 2.7.0).

For subsequent analyses, incidence angles of the laser beam are calculated for each TLS point. The incidence angle is defined as the angle between surface normal and incoming laser beam. Given the horizontal character of the study sites, the surface normal is assumed to be orthogonal to the horizontal plane for every point.

One critical point for quantifying ground movements in Arctic regions is to properly identify the actual ground surface. In

- tundra environments, this seemingly simple decision is complicated by the fact that the mineral soil is typically covered by a thick moss- lichen layer, as well as vascular plants. This moss-lichen surface is typically rather porous, and varies in height at very small scales. In this study, we define the ground surface as the solid moss-lichen surface that covers the underlying mineral layers. For excluding laser points that originate from and vascular vegetation that is above the ground surface, Bremer and Sass (2012) report that most surface-based vegetation filters do not achieve satisfying results for dense shrub vegetation.
- Thus, we follow the local minimum approach and extract ground points based on the criterion: lowest z-value within a defined neighborhood. Optimal search radius for identifying local neighbors is evaluated based on the GNSS ground measurements and is set to 2.5 cm. Subsidence is quantified by calculating ground surface elevation differences between two survey dates, assuming that the thickness of the moss-lichen layer does not change significantly
- To analyze the effect of vegetation on the deformation analysis, we calculate different measures to describe vegetation characteristics, such as range of z-values per pixel cell and RMS of z-values. Micro-topographic characteristics are considered by extracting local depressions within the study sites. Local depressions are identified by selecting the local minimum within a 2D search radius of 0.5 m.

#### 3.4 Vertical deformation analysis

For quantifying vertical movements of the ground surface that occur between the TLS surveys, we compare DEM differencing as a raster-based approach (e.g. Abellán et al. (2009), Goodwin et al. (2016)) with a point-based approach (Lague et al., 2013). The existing workflows are extended by a selection step that aims at identifying suitable ground surface points for the deformation analysis. Thereby we build on the finding that elevation errors due to (short) ground vegetation can be reduced

by capturing the area from multiple scan positions (Fan et al., 2014). To get further insights and validate our method, deformation analysis is performed based on (1) all ground surface TLS points and (2) after selecting those ground surface TLS points with neighboring points from other scan positions, i.e. locations that have been seen from multiple scan positions.

## 3.4.1 DEM differencing

In this study, firstly, DEMs are generated based on the lowest z-value within each raster cell and evaluated using the GNSS ground measurements. Within this comparison different TLS raster cell sizes (from 1 cm up to 500 cm) are evaluated to find the best compromise between capturing small-scale topographic variabilities and removing non-ground points from the TLS datasets. Secondly, we assess the quality of raster-based subsidence monitoring using difference DEMs.

## 3.4.2 M3C2 distance calculation

- Compared to the raster-based approach, point-based approaches for deformation analyses incorporate the full 3D information. We use the M3C2 distance calculation algorithm presented by Lague et al. (2013) to determine ground movements between the three survey dates. Main advantage of this method is that it accounts for position uncertainties, i.e. random errors due to averaging of point positions (Barnhart and Crosby, 2013). The M3C2 algorithm consist of two parts: Firstly, point normal estimation is performed and secondly, differences are calculated. For performing the M3C2 analysis, we fixed the normal in
- vertical direction, as the focus of this study in on vertical ground surface changes. The parameter projection scale set to 0.1 m, and the maximum displacement distance to 1.0 m. Significance of change rates is calculated by taking the (co-)registration error and local surface roughness into account (Lague et al., 2013). According to Abellan et al. (2016), the M3C2 algorithm has gained wide acceptance and Barnhart and Crosby (2013) state that it is the better tool (compared to e.g. direct cloud to cloud comparison) to resolve small changes.

#### 20 3.4.3 Point filter strategies

To account for sampling effects, two filter strategies are introduced that are applied to the TLS point clouds of each survey period before M3C2 distances are calculated. We aim at identifying positions within the point cloud that are "visible" from more than one scan position. Visibility is met if within a defined threshold around one position, a TLS point from another scan position can be found. The 3D search distance (r<sub>sp</sub>) for the visibility check is set to 0.5 cm to account for TLS position

- uncertainties. We investigate and compare two different filter strategies, which are the following (Fig. 3):
  - (1) Select TLS point from TLS ground surface points if visible from multiple scan positions.
  - (2) Select TLS point from all TLS points (and not from the TLS ground surface point only) if visible from multiple scan position. Then, select the local minima from all visible TLS points (within an infinite vertical cylinder, r<sub>min</sub>=2.5 cm). Thus, with strategy 2, firstly, all positions are selected that are visible from more than one scan position. Secondly, the local
- minima of those points are extracted. We expect filter strategy 1 to be the more rigorous criterion resulting in a reduced number of points available for the deformation analysis. In contrast, strategy 2 retains more TLS points, but deformation rates might


be less reliable. Once the relevant points are selected in each TLS dataset, the M3C2 distances are calculated to quantify the vertical movements.

The search radius  $r_{min}$  is set to 2.5 cm, considering the GNSS-based DEM validation (Section 4.1.1 DEM accuracy assessment for details). The 3D search radius  $r_{sp}$  is set to 1 cm. To account for position uncertainties of the TLS a threshold of 0.5 cm is chosen to select the local minima (i.e. the lowest point within a vertical search cylinder is selected as well as all points with z-

distance <0.5 cm to the lowest point).

#### 3.5 Quality assessment

Standard measures for comparing laser scanning data with a validation dataset (e.g. GNSS measurements) are, for example, the root-mean-square error (RMSE), the mean error and the standard deviation (std.) of error. Furthermore, we calculate robust

accuracy measures, which take outlier values and non-normal error distributions into account (i.e. median, normalized median absolute deviation [NMAD], 68.3% and 95% quantile) (Hämmerle et al., 2016; Höhle and Höhle, 2009).

#### 4 Results and discussion

## 4.1 Raster-based deformation analysis

#### 4.1.1 DEM accuracy assessment

The derived DEMs are validated using the GNSS ground measurements. We test cell sizes between 1 and 500 cm and find a resolution of 5 cm to be the best compromise between capturing small-scale topographic variabilities and removing non-ground points from the TLS datasets (Figure 4, and Appendix 2): The median error of height differences between TLS and GNSS is 0.1 cm up to 0.8 cm for the three survey periods (cell size = 5 cm). The robust estimator of standard deviation, the NMAD, ranges between 0.8 and 4.1 cm and the RMSE varies between 1.5 and 4.3 cm (Appendix 3). For small cell sizes (< 5 cm) the TLS elevation values are higher than those measured with the GNSS.</p>

## 4.1.2 Difference DEMs


Area-wide difference rasters are obtained by subtracting the TLS DEMs of two survey dates (cell size = 5 cm). The average TLS-based vertical change rate at site 1 for the periods 2015/06-2015/08 and 2015/06-2016/08 is about -2 cm (std. = 7 cm), and at mm-level (std. = 4 cm) between 2015/08 and 2016/08. At site 2 the extracted values suggest that the seasonal subsidence rates are lower compared to site 1 (about 1 cm between 2015/06-2015/08, std. = 7 cm).

However, in the difference maps the effect of the scanning setup on the multi-temporal comparison becomes visible (Figure 5). For the comparison between 2015/06-2015/08 and 2015/06-2016/08 (i.e. data acquisition is repeated from different scan positions) negative change rates next to the scan positions of August 2015 and August 2016, respectively, seem to be an overestimation of change. Next to scan positions of June 2015 the opposite effect is detected, which would indicate that change

5

rates are positive (i.e. an artificial heave is detected). Following our hypothesis in correspondence with Fan et al. (2014), this is caused due to the comparison of areas that have been captured at different incidence angles, meaning that the sampling differs between the two surveys (Fan et al. 2014). If the area is captured from similar scan positions (2015/08-2016/08), this effect is less dominant as shown in Figure 5(d). Nevertheless, it is noticeable that the difference map shows higher change rates (on cm-level) next to the scan positions compared to areas at further distance.

Further in-situ measurements are considered for evaluating the results. Compared to measurements at the subsidence stations, TLS-based difference DEMs tend to show smaller subsidence rates (median error between TLS and subsidence stations up to -1.2 cm, Appendix 4). All in all, this reveals a significant limitation of TLS DEM differencing for detecting spatial patterns of small-scale subsidence.

#### 10 4.2 Point-based deformation analysis

#### 4.2.1 Vertical changes

Vertical changes between point clouds of two surveys are derived using the M3C2 distance calculation. Figure 6 shows the resulting change maps for site 1 (seasonal subsidence rates, survey 2015/06 to 2015/08) based on (1) all TLS<sub>min</sub> points, (2) TLS<sub>min</sub> points that are illuminated by more than one scan position (strategy 1) and (3) local minima of those TLS points that

- 15 are visible from at least two scan positions (strategy 2). For a map displaying the changes for site 2 see Appendix 5. Whereas the sampling effect is visible when no selection is applied, filter strategy 1 masks out the affected areas and reduces the total number of difference values (8% of the all TLS<sub>min</sub> points, total number of points with strategy 1: 537,728). With filter strategy 2 the number of points representing vertical changes is 53% of the unselected dataset (total number: 3,532,408). Figure 6(c) visualizes that the sampling effect is less dominant when applying point selection. It needs to be noted that regarding the mean
- and median of change the filter strategies differ at mm-scale only. However, standard deviation and value range of changes is reduced by more than 50% with filter strategy 1. We confirm these findings by plotting change rate against incidence angle in Figure 7. While the incidence angle effects change rates if the deformation analysis relies on the entire TLS<sub>min</sub> dataset, applying filter strategy 1 reduces the sampling effect. However, with higher incidence angles i.e. larger distance to the scan positions, a wider scattering of the change rates occur. A larger (vegetation) error is also reported by Fan et al. (2014) with increasing scan
- 25 distance (and incidence angle). In this study within the pre-processing we filter the TLS dataset to a maximum distance of 30 m to the scan positions. A severer distance or incidence angle filter would allow reducing more noise (Fig. 7: filter strategy 1), but is a trade-off against number of available TLS points and, thus, the total area for which change rates can be derived. Taking the difference maps into account confirms that the applied filter strategies are suitable for reducing the detected overand underestimation of change rates and, thus, the observed sampling effects. In summary, the findings lead to the conclusion
- 30 that the selection of TLS points that are visible from multiple scan positions allows for revealing small-scale spatial change patterns that are not influenced by sampling effects.

The result presented above is confirmed by analyzing the changes within site 2 and, further, by taking into account the survey period of 2015/06 to 2016/08 as additional reference (Table 2): The statistics show that standard deviation of differences is reduced by 22% up to 58% for filter strategy 1 and by 10% up to 25% for strategy 2. On plot-level the median subsidence rates are consistent for the three survey periods at both sides. Change rates at site 2 tend to be lower (on mm-scale), whereas standard

deviation of differences with point selection is higher (up to 1.4 cm) compared to site 1. This can indicate an influence of sitespecific characteristics (e.g. vegetation height) on the deformation analysis.

By comparing TLS point clouds captured in August 2015 and 2016, an annual median subsidence rate of -0.4 cm is detected. TLS data were captured from similar scan positions according to scan setup 2. Again, both filter strategies are applied to both TLS point clouds: Filter strategy 1 and 2 reduce the number of difference values by a similar percentage as with scan setup 1

(down to approx. 10% with filter strategy 1, and approx. 50 % with strategy 2). Furthermore, the 95<sup>th</sup> percentile is lower compared to performing the comparison analysis based on all TLS<sub>min</sub> points.
 The visual evaluation of the resulting change maps for site 1, survey period 2015/08-2016/08 (Appendix 6) confirms that areas

showing high subsidence or heave rates are masked out by filter strategy 1 and partly also by strategy 2. Based on the presented results we conclude that strategy 1 leads to a more rigorous selection compared to strategy 2 i.e. less points are selected by

strategy 1 compared to strategy 2. Both approaches reduce the sampling effect in the multi-temporal deformation analyses, not only if data acquisition is repeated from different but also from the same scan position.

#### 4.2.2 Comparison with manual subsidence stations

TLS-based change rates are compared with in-situ measurements gathered manually at the subsidence stations. These stations provide 12 change measurements per site. For comparison, more than 6 million data points are retrieved from the TLS datasets

- (leading to about 500,000 difference measurements when applying the more rigorous filter strategy 1). For each subsidence station the closest TLS change rate is extracted. Differences between the change rates at the subsidence stations are presented in Figure 8 for site 1 (Appendix 7 for site 2). For the survey period 2015/06-2015/08, the median and the RMSE is reduced by applying the filter strategies. This finding cannot be confirmed for the survey periods 2015/06-2016/08 and 2015/08-2016/08. For those survey periods, manual measurements using subsidence stations show higher subsidence rates. Furthermore,
- applying the filter strategy does not decrease the RMSE. One explanation could be uncertainties that might be introduced when measuring subsidence manually at subsidence stations (see section Study area, field measurements and sensors for more details). Another point is the spatial sampling of the subsidence stations, which are placed in areas of sparse vegetation that are visible from various scan positions and, thus, being little affected by occlusion effects. Further research is needed to better understand the uncertainties of manually measured subsidence rates and its implication of methodological differences in
- monitoring subsidence compared to TLS or other remote sensing techniques.

## 4.2.3 Effect of vegetation and micro-topography

Fan et al. (2014) describe errors that occur in TLS ground measurements due to short vegetation. Another major concern in TLS deformation analysis are uncertainties due to surface roughness (Barnhart and Crosby, 2013). In the following, we analyze the effect of the presented point selection workflow on areas that are considered to influence TLS-derived change rates due to

- 5 their vegetation and micro-topographic characteristics (e.g. dense/high vegetation). This is done exemplarily for site 1, survey period 2015/06-2015/08. Figure 9 shows site 1 and the areas selected by filter strategy 1, local depressions and vegetation heights. Whereas without point selection the percentage of pixels representing local depressions is 20%, the percentage is reduced to 10% by applying filter strategy 1. When looking at the distribution of vegetation heights (i.e. range of z-values) within areas selected by filter strategy 1, we find a medium height of 9.5 cm compared to 13 cm in areas that are masked out.
- Standard deviation of vegetation heights is 7.8 cm in the included areas compared to 18 cm in the excluded areas (Appendix 8 for more details). These findings, firstly, confirm that vegetation does have an influence on TLS-based subsidence monitoring. This effect tends to increase in areas covered by higher and/or denser vegetation which aligns with e.g. Fan et al. (2014). Likewise in local depressions the sampling effects tends to be more dominant. Secondly, we show that the presented filter strategies can be used to mask out such areas from the deformation analysis by taking indirectly the scan setup into account
- 15 and without requiring any a priori knowledge.

#### **5** Conclusions



In this study we show that if the ground surface is captured by more than one TLS scan position, plausible subsidence rates (up to mm-scale) can be derived; compared to e.g. standard raster-based DEM difference maps which contain change rates strongly affected by sampling and occlusion effects. While those effects emerge in the standard difference maps, the average subsidence rates on plot-level are consistent with the applied calculation methods: seasonal subsidence is about -2 cm

- (2015/06-2015/08) and approx. -0.5 cm over one entire year (2015/08-2016/08) at our two study sites.
  Overall, the limitations of DEM differencing and point-based distance calculation based on TLS for monitoring small-scale vertical movements are presented. This effect has special relevance in Arctic tundra ecosystems due to a dense moss-lichen layer, which covers the underlying organic and soil layers, and, against this background, occurs even if data acquisition is
- 25 repeated from similar scan positions. Besides spatial sampling and occlusion effects, position and co-registration uncertainties influence quantification of small-scale changes.

Accordingly, we recommend the following to maximize the accuracy of TLS based subsidence monitoring:

(i) Co-registration is an important prerequisite to enable quantification of vertical changes on mm/cm-scale. Systematic shifts between multi-temporal TLS datasets can be reduced based on permanent fix points in an otherwise highly dynamic permafrost environment, for example, by applying a rigid 3D transformation.