# Peer review of "Terrestrial laser scanning for quantifying small-scale vertical movements of the ground surface in Artic permafrost regions"

_Earth Surface Dynamics, 2017_

## Referee Comment (RC1) · Anonymous Referee #1 · 16 Aug 2017

General comments:

This paper addresses an important and evolving topic of using terrestrial laser scanning for studying Artic permafrost. However, I have several questions and suggestions which need to be addressed. I am detailing them below. Nevertheless, the intent and objective behind this work presently are unclear to me and need to be addressed properly. If this work is about methodology development, then I must say that there is very less novelty involved and the advancements in the preexisting approaches are not significant. If this work is about reporting the state of changing permafrost in the Arctic then that part is feebly touched upon, understandably because the changes over a pe-

riod of just one year would not be as drastic as to definitely comment on the state of the permafrost. It needs more observations. Presently, I feel that although this is a good initiative, this is also a premature reporting and the authors can make a better study by gathering data from more sampling sites and for different seasons over duration of 2-3 years to report on the seasonal dynamics. The highlights of this research need to be emphasized in the present version.

Specific comments:

1. Entire paper needs a thorough language editing in the sentence structures and presentation styles. There is a lot of redundancy that can be avoided.

2. Abstract: It needs modification, particularly in the conclusive lines. The relevance of the research has been mentioned through the four opening lines. It can be shortened. 2-3 more sentences can be added elaborating more on the results, particularly mentioning some quantitative assessments. The end is abrupt and it can be modified by adding a conclusive line that highlights the contribution of the study in filling the research gaps and the future prospects.

3. Introduction:

P1 L23: Add reference for the first line of the introduction.

P2 L3: "...allow a detailed..." Replace "detailed" by "spatially continuous".

P2 L25-27: "Such ALS...cm". The resolutions have improved in recent years, particularly with the use of UAV-mounted ALS. So this sentence needs improvement.

4. Study area:

P4 L10-11: Is it the annual average temperature or an average from 1971-2000?

P4 L12-13: How far are the 2 sites from each other? In Figure 1, they appear to be ∼100 m apart. As mentioned "Site 1 is about 50x40 m, almost flat and covered by low tundra vegetation and Site 2 is equal in size but contains more shrubs", I was

wondering that the sites are so close that they must be having overlapping areas (going by the 50X40 m dimensions and considering that the dots in Fig.1 represent the middle point of the plots). What was the need to keep the sites so close? Is there really any difference in the vegetation as it seems similar for both the sites in Figure 1? I would suggest displaying close-range photos to establish it. These photos should not be only for a single time period for each of the sites but must be consisting of repeat images of the same sites during various surveys so that the reader can visualize the changes in the vegetation.

The legends from fig. 1 are missing. What do the colors in the map represent?

I do not like the fact that although the paper is about a permafrost region, there is not enough description of that in the study area section. Mere citation of a few articles is not enough. The authors must talk about the MAAT and monthly temperatures through some graphs using the data from the nearest monitoring station for past several years to show that the region can still be considered as of having a continuous permafrost. Several close range photos of relevant surface features corroborating the permafrost (for e.g., palsa or hummocks) in the study area could have been added here. This section needs improvement and justifications.

P4, L24-27: "Additionally, . . . environment." This is an extremely important step for repeat surveys. I would have liked to see the pictures of the installed rods. Were these rods marked? How exactly did they serve the purpose of common reference points during the repeat surveys? Did the authors also check for the change in the inclination of the rods during repeat surveys? These points need detailing because it's just a matter of mm scale accuracy and any error in the methodology can compromise the entire results. Presently, I cannot comment on the accuracy standards unless I get the information on this step.

5. Methods:

P6, L9: Full form of OPALS?

There is no description of interpolation and DEM raster generation algorithms. This cannot be avoided.

6. Results:

P8, L19: NMAD and RMSE are quite high for a TLS-based DEM! Figure 4 represents the poor accuracy of the DEMs.

P8, L23: "change rate" is actually absolute change. How can it be called "rate"?

7. Conclusions:

P11, L20-21: Why is the seasonal subsidence (just 2 months gap) more than the yearly subsidence? If seasonal subsidence is in cm then the yearly subsidence should also be at least in cm and not in mm. The conclusion seems to be consisting of several general points such as multi-point scanning for increasing the accuracy. However, what is seriously lacking is a clear-cut advancement provided by the present research and precise future prospects for studying the Arctic permafrost.

The levels of accuracy achieved during the entire field data collection (table 1, appendix 1) does not seem sufficient to me for commenting on the validity of the results.

---

## Referee Comment (RC2) · Anonymous Referee #2 · 18 Oct 2017

The manuscript makes an attempt to illustrate and evaluate a methodology for permafrost surface morphological change assessment using repeat TLS surface mapping. This is a worthwhile and long-awaited theme of research and is relevant for readers of Earth Surface Dynamics. As such, publication is warranted as this will undoubtedly be an area of highly active research moving forwards. However, I believe this paper, in current form, is not worthy of publication as I believe it does not meet basic criteria either for a proven innovation in methodology or new information about rates of change over a permafrost landscape. Indeed, upon starting to read the ms, the reader has high expectations for a demonstrated effective new methodology or about spatial variation in permafrost land surface change patterns but those expectations were not met. TLS

surface change using the methods advocated is not in-itself new and - more importantly - the conclusions present recommendations which experts in the field of TLS change detection might consider trivial and a priori self-evident. Regarding the highly anticipated permafrost dynamic component, we are told merely that the observed changes are "plausible" but the changes are weakly validated, despite the descriptions of control point arrays and field measurements. Given the 'observed' changes are not related with confidence to driving mechanisms and that they are close to noise levels in the analysis, we don't learn much that is new and it does not follow that a "plausible" rate of change is proof that change has been observed. Of note, one area of uncertainty that is not adequately addressed is that of changing moisture condition influences in apparent seasonal or inter-annual surface heave or subsidence.

I have not rejected this paper, however, as I believe documentation of the experiment is worthwhile given so few TLS data collections have been performed with a view to quantifying change in these landscapes. The application obviously has merit and should be further refined / evaluated. However, I would prefer to see a more thorough analysis with the current conclusions downplayed to match the evidence presented by the data. i.e. avoid implying that "plausible" rates of change are evidence that change has been observed. This is not proof. Also, I was a little confused as to why there are so many figures in appendices. In some cases, the cited figures would be logically embedded, while in others it is clear the figures are not needed. I find this aspect of presentation cluttered and confused. Finally, while referencing is adequate, I find there are instances I am aware of where the paper does not cite original studies but rather focusses on recent studies that have recreated the innovations of others. Alas, this seems to be a trend in published material these days so is not a 'rejectable' offence but it does suggest the authors could have done more homework on their topic of study.

My specific comments are not exhaustive, as I think the manuscript needs to be re-framed as a technical evaluation as opposed to demonstration, and the recommendations need to go beyond the obvious and a priori self-evident into actionable criteria on

sampling and analytical design (I make some suggestions below). I also believe the criteria for success in the experiment need to be defined. If this is done, however, I believe the experiment will fail to achieve its goal. That, however, does not preclude publication rather it does provide an opportunity for the authors to make recommendations over what is needed to achieve success? For example, it may be that collection or processing improvements could be made or that there are hard limits on the magnitude of change that can be observed with confidence under certain circumstances. Detailing these technical and circumstantial constraints would be informative for future studies. Stating results are plausible does little to advance the science or our understanding.

Specific comments:

P1 L19-21 - here and elsewhere I find that the text is suggesting two mutually exclusive concepts (i.e. multiple scan positions vs raster DEM differences) when in fact one concept could be a subset of the other. This needs to be tightened up throughout, as the implication is that these are distinct methods (which I assume they are in the way executed here) but in fact raster approaches could be applied either to single or multiple scans so the distinction the author is aiming for is not obvious.

P2 L11-17 – In this section, it is implied (though not stated) that ALS assessment of permafrost-related processes and associated surface and vegetation dynamics has not yet been evaluated / published. However, I'd like to draw the authors' attention to a recent paper by Chasmer et al (2017) in Global Change Biology ("Threshold loss….."). Here permafrost loss and associated vegetation dynamics were explicitly evaluated and rates of change quantified using time series ALS.

P9 L29-31 – Two points. i) I'm not sure the evidence presented can be taken as proof that any spatial change has been observed with confidence at all. I think this is a general conclusion that requires further proof; ii) it is a priori self-evident that multiple samples or measurements of the same object will tend to increase confidence in the observation being made. As such, I'm not sure why much is being made of this

'finding'. Many TLS papers going back over a decade that deal with vegetation and other surface attribute sampling and modeling have either implicitly made this assumption without needing to prove it or have directly observed that multiple scan sampling improves results vs single scan sampling. On the one hand, I think this result is not worthy of recording because it is trivial but on the other, if the authors' feel it is worth commenting upon, then I suggest placing this observation into the context of existing TLS plot sampling literature.

P10 L5 – "This can indicate….." I'm not sure what to make of this statement. Does it mean that surface deformation is influenced by vegetation attributes? If so, this would be a worthwhile topic to further explore and characterise. Stating that something might be happening is not instructive but if it is (or even might be) happening then providing further evidence and discussion would be worthwhile given the stated topic of the paper and journal.

P10 L7 – i) minor point but is a negative subsidence heave? Here we see a subsidence rate of -0.4cm. I think this needs to be clarified to avoid confusion. ii) more importantly, however, could not 4mm of change simply be due to differences in moisture content in the sphagnum between data collections? I am not familiar with the depth or volume of moss overlying the TVC study site but in areas to the south I do know peat moss surface scan expand and contract vertically by up to 5cm in a single season. As such, given we are here seeing changes that are generally less than 2cm and even discussing differences at mm-level, how much of this change could be due to moisture differences in the surface organics? This may not have been studied directly but it at least must be acknowledged as a source of uncertainty or discounted using an evidence-based argument.

P11 L11 – it is ok to confirm that TLS subsidence monitoring will be influenced by vegetation. However, this is an odd 'finding' as – again – it is a priori expected to be true without the need for verification. What would be more instructive, in this regard, would be to quantify to what degree certain types or heights or densities of vegetation

influence rates of observed subsidence. The text and Figure 9 hint that this might be possible but the analysis does not get far enough into such stratification. As such, the stated finding is trivial and does not enhance our understanding.

P11 L17 – as already mentioned, noting that the observed surface differences match plausible rates of subsidence is not an informative result; especially when those rates are close to noise-level in the analysis anyway….and when other influences like changes in surface moisture and sphagnum expansion / contraction have not been considered.

P11/12 L27 + Recommendations: (i) I don't see co-registration of scans as an appropriate recommendation of this study. Indeed, it is a priori required to co-register scans for this type of analysis. The specific aspect of requiring permanent fixed reference points is appropriate, though also to be expected given the existing literature on change detection over dynamic surfaces. If the recommendation were to provide specifics on design elements for TLS change assessment over an actively subsiding permafrost surface that are based on findings of this study, then that would be instructive given the authors' somewhat unique experience in this specific experimental context. As such, more in-depth analysis of the rigidity of the ctrl marker array through time - and where outliers may have occurred - would be informative. (ii) It seems obvious that spatial sampling, surface undulation and vegetation cover will influence the accuracy of deformation analysis as a result of specific location-based occlusions. This should be expected, so to state this in the conclusion as part of a recommendation for spatial averaging seems redundant. A more quantified statement concerning under exactly which conditions spatial averaging is needed and to what degree would be more actionable for future experimental design. (iii) Not clear why it is worth recommending multiple scan positions. This is a priori known from previous TLS plot-sampling studies and it follows logically / intuitively that increased independent sampling will increase accuracy. However, it is instructive, for example to document errors associated with a single scan vs 2 scans vs multiple scans and from different geometric sampling

locations. Following such analysis, a specific recommendation regarding number of scans and the geometric configuration necessary to meet certain accuracy requirements could be made and would be very useful. (iv) Not immediately clear how this is a recommendation or how it functionally differs from the previous recommendation.

P12 L10 – I don't agree that the results – as presented – demonstrate that TLS provides highly accurate ground truth data for subsidence monitoring over the time or space scales evaluated at this particular study site. I do agree it has the potential to do so, however, but the statement needs to be adjusted so it is based on the actual provable findings of this study.

---

## Author Comment (AC1) · 14 Nov 2017

**Referee Comments 1**

*Dear reviewer, we would like to thank you for your helpful suggestions and constructive comments on our manuscript. They will help us to further improve and develop our work. We provide a response to your major points below in italics.*

General comments:

This paper addresses an important and evolving topic of using terrestrial laser scanning for studying Arctic permafrost. However, I have several questions and suggestions which need to be addressed. I am detailing them below. Nevertheless, the intent and objective behind this work presently are unclear to me and need to be addressed properly. If this work is about methodology development, then I must say that there is very less novelty involved and the advancements in the preexisting approaches are not significant. If this work is about reporting the state of changing permafrost in the Arctic then that part is feebly touched upon, understandably because the changes over a period of just one year would not be as drastic as to definitely comment on the state of the permafrost. It needs more observations. Presently, I feel that although this is a good initiative, this is also a premature reporting and the authors can make a better study by gathering data from more sampling sites and for different seasons over duration of 2-3 years to report on the seasonal dynamics. The highlights of this research need to be emphasized in the present version.

*Our manuscript presents a methodological study which evaluates TLS for quantifying small-scale thaw subsidence. The scientific contributions are that, firstly, we assess the applicability of current state-of-the-art approaches for TLS-based subsidence monitoring, which is challenging in Arctic tundra-ecosystems due to a typically dense moss-lichen layer and micro-topographic characteristics. We show, for example, that standard DEM differencing in Arctic tundra-ecosystems is error-prone due to spatial sampling effects. Secondly, we introduce a new point-based filter strategy to overcome the described spatial sampling and signal occlusion effects. The presented method identifies TLS ground points suitable for multi-temporal deformation analyses and allows to deliver highly accurate ground-truth data for small-scale subsidence. Finally, recommendations for TLS subsidence monitoring are given.*

Specific comments:

1.  Entire paper needs a thorough language editing in the sentence structures and presentation styles. There is a lot of redundancy that can be avoided.

    Abstract: It needs modification, particularly in the conclusive lines. The relevance of the research has been mentioned through the four opening lines. It can be shortened. 2-3 more sentences can be added elaborating more on the results, particularly mentioning some quantitative assessments. The end is abrupt and it can be modified by adding a conclusive line that highlights the contribution of the study in filling the research gaps and the future prospects.

2.  Introduction:

    P1 L23: Add reference for the first line of the introduction.

    *Zhang et al., 2008 and van Everdingen, 2005 are the references for this line:*

*Zhang, T., Barry, R. G., Knowles, K., Heginbottom, J. A., and Brown, J.: Statistics and characteristics of permafrost and ground-ice distribution in the Northern Hemisphere, Polar Geography, 31, 47–68, doi:10.1080/10889370802175895, 2008.*

*van Everdingen, R. O.: Multi-language glossary of permafrost and related ground-ice terms: http://globalcryospherewatch.org/reference/glossary.php.*

P2 L3: "...allow a detailed... " Replace "detailed" by "spatially continuous".

P2 L25-27: "Such ALS...cm". The resolutions have improved in recent years, particularly with the use of UAV-mounted ALS. So this sentence needs improvement.

*New technologies like Unmanned Laser Scanning (ULS) have been recently introduced, showing potential for mapping topography and vegetation with higher point density and accuracy (e.g. Wieser et al., 2016). Nevertheless, the registration accuracy of kinematic acquisition systems is still lower (up to cm level) compared to static data acquisition (up to mm level). We therefore rely on TLS for monitoring small-scale subsidence.*

3. Study area:

P4 L10-11: Is it the annual average temperature or an average from 1971-2000?

*It is the mean annual air temperature (MAAT) between 1971 and 2000.*

P4 L12-13: How far are the 2 sites from each other? In Figure 1, they appear to be ~100 m apart. As mentioned "Site 1 is about 50x40 m, almost flat and covered by low tundra vegetation and Site 2 is equal in size but contains more shrubs", I was wondering that the sites are so close that they must be having overlapping areas (going by the 50X40 m dimensions and considering that the dots in Fig.1 represent the middle point of the plots). What was the need to keep the sites so close? Is there really any difference in the vegetation as it seems similar for both the sites in Figure 1? I would suggest displaying close-range photos to establish it. These photos should not be only for a single time period for each of the sites but must be consisting of repeat images of the same sites during various surveys so that the reader can visualize the changes in the vegetation.

*The modified version of Figure 1 (see below) shows the actual ALS extents: The sites do not have an overlapping area. We agree that it is not easy to visually recognize differences in the vegetation based on the provided photo: Regarding the photo of site 1: The shrubby area next to the scanner is not part of the study site. We replaced this photo to provide a better impression how site 1 looks like. Furthermore, the photographer's position was marked in the map. Now it is visible that site 1 contains considerably less shrubs compared to site 2.*

The legends from fig. 1 are missing. What do the colors in the map represent?

*The orthophoto used as background map is a RGB image captured in August 2015.*

[Figure]

I do not like the fact that although the paper is about a permafrost region, there is not enough description of that in the study area section. Mere citation of a few articles is not enough. The authors must talk about the MAAT and monthly temperatures through some graphs using the data from the nearest monitoring station for past several years to show that the region can still be considered as of having a continuous permafrost. Several close range photos of relevant surface features corroborating the permafrost (for e.g., palsa or hummocks) in the study area could have been added here. This section needs improvement and justifications.

*We had a look at the meteorological data from a weather station at TVC (distance to the study sites: < 1km). The MAAT measured at TVC was -7 °C from 2013 to 2016. The regional permafrost is estimated to reach depths between 100 m and 150 m (Marsh et al. 2008) and maximum active layer depths ranging from 0.4 to 0.8 m (Qinton and Marsh 1999).*

*Marsh, P., Pomeroy, J., Pohl, S., Quinton, W., Onclin, C., Russell, M., Neumann, N., Pietroniro, A., Davison, B., & McCartney, S. (2008). Snowmelt Processes and Runoff at the Arctic Treeline: Ten Years of MAGS Research. In M.-k. Woo (Ed.), Cold Region Atmospheric and Hydrologic Studies: The Mackenzie GEWEX Experience (Vol. 2: Hydrologic Processes). Berlin, Heidelberg: Springer. pp. 97-123. doi: 10.1007/978-3-540-75136-6_6.*

*Quinton, W. L. & Marsh, P. (1999). A Conceptual Framework for Runoff Generation in a Permafrost Environment. Hydrological Processes, 13, pp. 2563-2581.*

P4, L24-27: "Additionally, ... environment." This is an extremely important step for repeat surveys. I would have liked to see the pictures of the installed rods. Were these rods marked? How exactly did they serve the purpose of common reference points during the repeat surveys? Did the authors also check for the change in the inclination of the rods during repeat surveys? These points need detailing because it's just a matter of mm scale accuracy and any error in the methodology can

compromise the entire results. Presently, I cannot comment on the accuracy standards unless I get the information on this step.

*We installed the rods for two purposes. Firstly, to obtain reference data for the TLS-based subsidence rates. Secondly, we identified the top of each rod in the TLS point clouds and used those coordinates as fix points to co-register the TLS datasets. We did check for potential change in the inclination of the rods during repeat surveys by calculating the 2D-distances between all extracted coordinates. The 2D-distance is constant for all three survey dates (difference between 2D-distances < 1mm) meaning that the rods are stable.*

4. Methods:

P6, L9: Full form of OPALS?

*Orientation and Processing of Airborne Laser Scanning*

There is no description of interpolation and DEM raster generation algorithms. This cannot be avoided.

*The DEM raster generation is explained in section 3.4.1 DEM differencing: manuscript "DEMs are generated based on the lowest z-value within each raster cell [...] different TLS raster cell sizes (from 1 cm up to 500 cm) are evaluated."*

5. Results:

P8, L19: NMAD and RMSE are quite high for a TLS-based DEM! Figure 4 represents the poor accuracy of the DEMs.

*In this section (4.1 Raster-based deformation analysis) we evaluate raster-based DEM differencing as one approach to derive subsidence rates. Yes, we found that the DEMs and the resulting DEMs difference maps are of poor accuracy. Therefore, we state "All in all, this reveals a significant limitation of TLS DEM differencing for detecting spatial patterns of small-scale subsidence" (P9, L8). Our conclusion is not to rely on raster-based approaches but to apply our proposed filter strategy and then to calculate changes directly in the point cloud e.g. using the M3C2 distance calculation algorithm.*

P8, L23: "change rate" is actually absolute change. How can it be called "rate"?

*Thanks for the hint. Correct is "The average TLS-based vertical change for site 1 was recorded at approximately -2 cm ...".*

6. Conclusions:

P11, L20-21: Why is the seasonal subsidence (just 2 months gap) more than the yearly subsidence? If seasonal subsidence is in cm then the yearly subsidence should also be at least in cm and not in mm. The conclusion seems to be consisting of several general points such as multi-point scanning for increasing the accuracy. However, what is seriously lacking is a clear-cut advancement provided by the present research and precise future prospects for studying the Arctic permafrost. The levels of accuracy achieved during the entire field data collection (table 1, appendix 1) does not seem sufficient to me for commenting on the validity of the results

*In Table 1 we present RMSE and mean error at five control points per study sites. RMSE and mean error (z-distance) after co-registration are of relevance to assess uncertainties for the TLS-derived vertical change maps (as suggested by Orem and Pelletier, 2015). As the co-registration error in z-direction is between 0.2–0.3 cm, our results support the observation that the TLS data are sufficient to derive small-scale subsidence maps. Appendix 1 shows the registration error between the single scan positions for each survey date which is of minor relevance to assess uncertainties of the derived results.*

---

## Author Comment (AC2) · 14 Nov 2017

**Referee 2**

*Dear reviewer, we would like to thank you for your helpful suggestions and constructive comments on our manuscript. They will help us to further improve and develop our work. We provide a response to your major points below in italics.*

General comments:

The manuscript makes an attempt to illustrate and evaluate a methodology for permafrost surface morphological change assessment using repeat TLS surface mapping. This is a worthwhile and long-awaited theme of research and is relevant for readers of Earth Surface Dynamics. As such, publication is warranted as this will undoubtedly be an area of highly active research moving forwards. However, I believe this paper, in current form, is not worthy of publication as I believe it does not meet basic criteria either for a proven innovation in methodology or new information about rates of change over a permafrost landscape. Indeed, upon starting to read the ms, the reader has high expectations for a demonstrated effective new methodology or about spatial variation in permafrost land surface change patterns but those expectations were not met. TLS surface change using the methods advocated is not in-itself new and - more importantly - the conclusions present recommendations which experts in the field of TLS change detection might consider trivial and a priori self-evident. Regarding the highly anticipated permafrost dynamic component, we are told merely that the observed changes are "plausible" but the changes are weakly validated, despite the descriptions of control point arrays and field measurements. Given the 'observed' changes are not related with confidence to driving mechanisms and that they are close to noise levels in the analysis, we don't learn much that is new and it does not follow that a "plausible" rate of change is proof that change has been observed. Of note, one area of uncertainty that is not adequately addressed is that of changing moisture condition influences in apparent seasonal or inter-annual surface heave or subsidence.

I have not rejected this paper, however, as I believe documentation of the experiment is worthwhile given so few TLS data collections have been performed with a view to quantifying change in these landscapes. The application obviously has merit and should be further refined / evaluated. However, I would prefer to see a more thorough analysis with the current conclusions downplayed to match the evidence presented by the data. i.e. avoid implying that "plausible" rates of change are evidence that change has been observed. This is not proof. Also, I was a little confused as to why there are so many figures in appendices. In some cases, the cited figures would be logically embedded, while in others it is clear the figures are not needed. I find this aspect of presentation cluttered and confused. Finally, while referencing is adequate, I find there are instances I am aware of where the paper does not cite original studies but rather focusses on recent studies that have recreated the innovations of others. Alas, this seems to be a trend in published material these days so is not a 'rejectable' offence but it does suggest the authors could have done more homework on their topic of study.

My specific comments are not exhaustive, as I think the manuscript needs to be reframed as a technical evaluation as opposed to demonstration, and the recommendations need to go beyond the obvious and a priori self-evident into actionable criteria on sampling and analytical design (I make some suggestions below). I also believe the criteria for success in the experiment need to be defined. If this

is done, however, I believe the experiment will fail to achieve its goal. That, however, does not preclude publication rather it does provide an opportunity for the authors to make recommendations over what is needed to achieve success? For example, it may be that collection or processing improvements could be made or that there are hard limits on the magnitude of change that can be observed with confidence under certain circumstances. Detailing these technical and circumstantial constraints would be informative for future studies. Stating results are plausible does little to advance the science or our understanding.

*Thanks for your suggestions – we will further improve our study and thereby focus on the technical challenges and potentials of TLS for monitoring small-scale changes in permafrost environments.*

Specific comments:

P1 L19-21 - here and elsewhere I find that the text is suggesting two mutually exclusive concepts (i.e. multiple scan positions vs raster DEM differences) when in fact one concept could be a subset of the other. This needs to be tightened up throughout, as the implication is that these are distinct methods (which I assume they are in the way executed here) but in fact raster approaches could be applied either to single or multiple scans so the distinction the author is aiming for is not obvious.

*We used the criterion "is visible from multiple scan positions" to filter the point cloud and to mask out areas that are strongly affected by sampling and occlusion effects. The remaining points are considered for the point-based deformation analysis. Indeed, we could also apply raster-based DEM differencing (instead of point-based distance calculation) to the filtered point cloud.*

P2 L11-17 – In this section, it is implied (though not stated) that ALS assessment of permafrost-related processes and associated surface and vegetation dynamics has not yet been evaluated / published. However, I'd like to draw the authors' attention to a recent paper by Chasmer et al (2017) in Global Change Biology ("Threshold loss. . ..."). Here permafrost loss and associated vegetation dynamics were explicitly evaluated and rates of change quantified using time series ALS.

*Thanks for this hint.*

P9 L29-31 – Two points. i) I'm not sure the evidence presented can be taken as proof that any spatial change has been observed with confidence at all. I think this is a general conclusion that requires further proof; ii) it is a priori self-evident that multiple samples or measurements of the same object will tend to increase confidence in the observation being made. As such, I'm not sure why much is being made of this 'finding'. Many TLS papers going back over a decade that deal with vegetation and other surface attribute sampling and modeling have either implicitly made this assumption without needing to prove it or have directly observed that multiple scan sampling improves results vs single scan sampling. On the one hand, I think this result is not worthy of recording because it is trivial but on the other, if the authors' feel it is worth commenting upon, then I suggest placing this observation into the context of existing TLS plot sampling literature.

*In the Introduction we refer to e.g. Fan et al. (2014) who state that scanning the same area from multiple locations reduces the vegetation error. We build on this finding and propose a filter strategy*

*which identifies areas sampled by multiple scan positions to increase confidence in the deformation analysis.*

P10 L5 – "This can indicate. . ..." I'm not sure what to make of this statement. Does it mean that surface deformation is influenced by vegetation attributes? If so, this would be a worthwhile topic to further explore and characterise. Stating that something might be happening is not instructive but if it is (or even might be) happening then providing further evidence and discussion would be worthwhile given the stated topic of the paper and journal.

*The influence of vegetation on the detected surface deformation is evaluated in the next section "4.2.3 Effect of vegetation and micro-topography". But yes, we agree – further in-depth analyses are needed to better understand the effect of low tundra vegetation (i.e. thick moss-lichen layer) on the TLS measurements and its ability to detect surface deformation.*

P10 L7 – i) minor point but is a negative subsidence heave? Here we see a subsidence rate of -0.4cm. I think this needs to be clarified to avoid confusion. ii) more importantly, however, could not 4mm of change simply be due to differences in moisture content in the sphagnum between data collections? I am not familiar with the depth or volume of moss overlying the TVC study site but in areas to the south I do know peat moss surface scan expand and contract vertically by up to 5cm in a single season. As such, given we are here seeing changes that are generally less than 2cm and even discussing differences at mm-level, how much of this change could be due to moisture differences in the surface organics? This may not have been studied directly but it at least must be acknowledged as a source of uncertainty or discounted using an evidence-based argument.

*Yes, differences in moisture content are a source of uncertainty. We will acknowledge this in future studies.*

P11 L11 – it is ok to confirm that TLS subsidence monitoring will be influenced by vegetation. However, this is an odd 'finding' as – again – it is a priori expected to be true without the need for verification. What would be more instructive, in this regard, would be to quantify to what degree certain types or heights or densities of vegetation influence rates of observed subsidence. The text and Figure 9 hint that this might be possible but the analysis does not get far enough into such stratification. As such, the stated finding is trivial and does not enhance our understanding.

*See response to previous comment (P10 L7).*

P11 L17 – as already mentioned, noting that the observed surface differences match plausible rates of subsidence is not an informative result; especially when those rates are close to noise-level in the analysis anyway. . ..and when other influences like changes in surface moisture and sphagnum expansion / contraction have not been considered.

*We will compare and evaluate the TLS measurements with other datasets in a future study. Adding this analysis to the presented intrinsic evaluation of our methods will give us further insights regarding constrains and benefits of TLS for subsidence monitoring.*

P11/12 L27 + Recommendations: (i) I don't see co-registration of scans as an appropriate recommendation of this study. Indeed, it is a priori required to co-register scans for this type of

analysis. The specific aspect of requiring permanent fixed reference points is appropriate, though also to be expected given the existing literature on change detection over dynamic surfaces. If the recommendation were to provide specifics on design elements for TLS change assessment over an actively subsiding permafrost surface that are based on findings of this study, then that would be instructive given the authors' somewhat unique experience in this specific experimental context. As such, more in-depth analysis of the rigidity of the ctrl marker array through time - and where outliers may have occurred - would be informative. (ii) It seems obvious that spatial sampling, surface undulation and vegetation cover will influence the accuracy of deformation analysis as a result of specific location-based occlusions. This should be expected, so to state this in the conclusion as part of a recommendation for spatial averaging seems redundant. A more quantified statement concerning under exactly which conditions spatial averaging is needed and to what degree would be more actionable for future experimental design. (iii) Not clear why it is worth recommending multiple scan positions. This is a priori known from previous TLS plot-sampling studies and it follows logically / intuitively that increased independent sampling will increase accuracy. However, it is instructive, for example to document errors associated with a single scan vs 2 scans vs multiple scans and from different geometric sampling locations. Following such analysis, a specific recommendation regarding number of scans and the geometric configuration necessary to meet certain accuracy requirements could be made and would be very useful. (iv) Not immediately clear how this is a recommendation or how it functionally differs from the previous recommendation.

*Thanks for your recommendations – we will try to address them in future studies.*

P12 L10 – I don't agree that the results – as presented – demonstrate that TLS provides highly accurate ground truth data for subsidence monitoring over the time or space scales evaluated at this particular study site. I do agree it has the potential to do so, however, but the statement needs to be adjusted so it is based on the actual provable findings of this study.

*In a following study we will consider further validation datasets to verify the presented findings and highlight more clearly the potential and challenges of TLS for subsidence monitoring.*